# Effect of retinoic acid treatment on the retinoic acid signaling pathway in a human siRNA-based aniridia limbal epithelial cell model, *in vitro*

Shao-Lun Hsu[ID][1]*, Tanja Stachon[1], Fabian N. Fries[1,2], Zhen Li[1], Shuailin Li[ID][1], Shanhe Liu[1], Berthold Seitz[ID][2], Swarnali Kundu[1], Maryam Amini[1], Shweta Suiwal[1], Nóra Szentmáry[1]

**1** Dr. Rolf M. Schwiete Center for Limbal Stem Cell and Congenital Aniridia Research, Saarland University, Homburg/Saar, Germany, **2** Department of Ophthalmology, Saarland University Medical Center, Homburg/Saar, Germany

* arinahsu@gmail.com

## Abstract

### Purpose

In the present study, we evaluate gene and protein expression levels in an *in vitro* siRNA-mediated PAX6 knockdown limbal epithelial cell (LEC) model after RA treatment. This study aims to investigate the direct effects of active RA products and their association with key regulators of the RA signaling pathway in siRNA PAX6 knockdown LECs, providing further insights into the potential role of RA signaling in AAK pathogenesis.

### Methods

Primary human limbal epithelial cells (LECs) were subjected to siRNA-mediated PAX6 knockdown to mimic PAX6 deletion in congenital aniridia (n = 8). Following knockdown, 0 μM, 1 μM, and 5 μM all-trans retinoic acid (RA) treatments were applied to both the siRNA PAX6 control and knockdown groups. After 48 hours of incubation, the mRNA expression levels of paired box 6 (PAX6), alcohol dehydrogenase 7 (ADH7), retinol dehydrogenase 10 (RDH10), aldehyde dehydrogenase 1 family member A1 (ALDH1A1), cytochrome P450 family 26 subfamily A member 1 (CYP26A1), retinol-binding protein 1 (RBP1), cellular retinoic acid-binding protein 2 (CRABP2), fatty acid-binding protein 5 (FABP5), retinoid X receptor alpha (RXRA), retinoid X receptor beta (RXRB), retinoic acid receptor alpha (RARA), retinoic acid receptor beta (RARB), peroxisome proliferator-activated receptor gamma (PPARG), and vascular endothelial growth factor A (VEGFA) were analyzed using qPCR. Protein expression levels were assessed using ELISA or Western blot, while cell proliferation rates were measured using the BrdU assay.

**Data availability statement:** All relevant data are within the manuscript and its Supporting information files.

**Funding:** The work of Shao-Lun Hsu, Tanja Stachon, Fabian N. Fries, Zhen Li, Shuailin Li, Shanhe Liu, Swarnali Kundu, Maryam Amini, Shweta Suiwal, Nóra Szentmáry at the Dr. Rolf M. Schwiete Center for Limbal Stem Cell and Aniridia Research was supported by the Dr. Rolf M. Schwiete Foundation (project number 08/2017). The author(s) received no specific funding for this work.

**Competing interests:** The authors have declared that no competing interests exist.

## Results

*PAX6, ADH7, ALDH1A1, FABP5* mRNA levels and PAX6, ADH7, ALDH1A1, FABP5, PPARG2, RARB protein levels were significantly lower in the PAX6 knockdown group, than in controls (p ≤ 0.018). *PPARG* mRNA level was significantly higher in the PAX6 knockdown group than in controls (p = 0.012).

*ALDH1A1* mRNA expression was significantly downregulated using 5 µM RA treatment in the control group (p = 0.038). *CYP26A1* mRNA expression was upregulated using 1 µM and 5 µM RA treatment in both the PAX6 control (p < 0.001; p < 0.001) and the PAX6 knockdown group (p = 0.001; p = 0.002). *CRABP2* mRNA expression in the PAX6 knockdown group (p = 0.02) and CRABP2 protein expression in both groups were downregulated using 5 µM RA concentration (p = 0.003; p = 0.02). Protein expression of RXRA was downregulated to 5 µM RA treatment in the controls (p = 0.007). mRNA expression of *RARA* in the PAX6 knockdown groups (p = 0.023) and mRNA expression of *RARB* in both groups (p = 0.007, p < 0.001) were downregulated to 5 µM RA treatment. RARB protein expression was downregulated to 1 µM and 5 µM RA treatment (p = 0.02, p = 0.004) in the controls. *VEGFA* mRNA expression in PAX6 controls was upregulated using 5 µM RA (p = 0.041).

Cell proliferation rate was downregulated in PAX6 knockdown groups compared to the controls and downregulated using 5 µM RA concentration only in the controls (p < 0.001, p = 0.025).

## Conclusions

Our results reveal a reduced proliferation rate in PAX6 knockdown LECs, along with a less pronounced downregulation of proliferation in response to increased RA concentration. Additionally, the study highlights altered expression of key regulators in the RA signaling pathway, influenced by both PAX6 activity and RA treatment. These findings suggest a potential disruption in RA-mediated cellular regulation in PAX6-deficient LECs.

## 1 Introduction

Congenital PAX6-aniridia is a rare panocular disease characterized structurally by dysgenesis of the outer segment of the eye and pathophysiologically by dysregulation of epithelial homeostasis, including impaired epithelial renewal, disrupted cell cycle control, and defective limbal stem cell maintenance [1,2]. Clinically, the condition is marked by the absence of the iris, as suggested by its name, along with widespread involvement of surface ectoderm-derived structures. This includes dysgenesis of the lens, trabecular meshwork, macula, and optic nerve to varying degrees, leading to the development of cataracts, secondary glaucoma, fundal coloboma, and progressive visual impairment [1,3,4]. A study by Enwright & Grainger

[5], demonstrated that lens formation fails to respond to retinoic acid (RA) during embryonic stage E9 in PAX6 knockout mice, highlighting the critical role of PAX6 in ocular development via the RA signaling pathway. Aniridia-associated keratopathy (AAK) affects 90% of congenital aniridia patients [6]. AAK is characterized by recurrent epithelial defects, delayed wound healing, fibrosis, limbal stem cell deficiency, conjunctivalization, and neovascularization of the limbus—pathological changes that are closely linked to PAX6 and its RA-mediated role in cell proliferation [7]. Furthermore, a study by Li et al. [8] found that corneal pannus tissues from patients with ocular surface diseases such as aniridia and recurrent pterygium exhibited reduced or absent PAX6 expression. The pathological process observed in these tissues resembled squamous metaplasia in adult rabbit central corneal transit-amplifying cells, ultimately leading to epidermal dysplasia and uncontrolled fibrovascular tissue overgrowth [8].

To identify key markers associated with PAX6-congenital aniridia, primary conjunctival epithelial cells from PAX6-AAK patients were screened in our previous study at both the mRNA and protein expression levels, compared to healthy controls [9,10]. The results suggest that some of the most significantly altered targets in gene and protein expression are regulators involved in the RA signaling pathway. These include RBP1 (retinol-binding protein 1), ADH7 (all-trans-retinol dehydrogenase 7), RDH10 (retinol dehydrogenase 10), ALDH1A1 (retinal dehydrogenase1), ALDH3A1 (aldehyde dehydrogenase 3 family member A1), CYP1B1 (cytochrome P450 1B1), CYP26A1 (cytochrome P450 26A1), FABP5 (fatty acid-binding protein 5), PPARG (peroxisome proliferator-activated receptor gamma), and CRABP2 (cellular retinoic acid-binding protein 2) [9,10]. To further investigate whether these gene expression changes were directly influenced by PAX6 or secondarily affected by AAK disease state, a PAX6 knockdown small interfering RNA (siRNA)-based model was established in healthy limbal epithelial cells (LECs) and compared to primary aniridia LECs [11]. The results demonstrated that ALDH1A1 and ADH7 mRNA levels were significantly reduced not only in aniridia limbal epithelial cells but also in the siRNA-PAX6 knockdown model [11]. This finding further underscores the essential role of PAX6 in regulating key enzymes within the RA signaling pathway.

Beyond PAX6, RA is a biologically active metabolite of retinol that plays a crucial role as a key regulator during embryonic ocular development in vertebrates [12]. RA signaling is essential for initiating the folding of the optic vesicle and the invagination of the lens placode during the 3rd to 5th week of gestation, followed by promoting the development of the ventral retina and optic nerve from the neural crest cell-derived periocular mesenchyme [13]. Studies have shown that the absence of RA at this stage leads to microphthalmia and corneal defects [14]. A simplified overview of the RA signaling pathway is illustrated in Fig 1. Retinol is first taken up through the cell membrane and undergoes a two-step oxidative process catalyzed by the metabolic enzymes ADH7 and ALDH1A1, converting retinol into functional RA. Once produced, RA can either be degraded by CYP26A1 or enter the nucleus to regulate the transcription of downstream genes [15,16]. Cellular RA-binding proteins, CRABP2 and FABP5, maintain a dynamic balance, regulating cellular functions through distinct pathways: CRABP2-bound RA activates the RXR/RAR transcription complex, while FABP5-bound RA interacts with the RXR/PPARG complex [17–19]. A study by Latta et al. investigated the expression levels of key regulators in the RA signaling pathway in LECs treated with retinol, RA, and RAR antagonists [20]. Their findings indicate that retinoid concentration and its derivatives influence the expression of multiple transcripts in the RA pathway, suggesting a potential dose-dependent regulatory mechanism. Moreover, studies suggest that RA concentration may drive cell proliferation by upregulating FABP5 expression [18,21,22]. An imbalance between FABP5 and CRABP2 expression has also been implicated in the pathogenesis of tumors and abnormal embryonic development in various tissues [18,22,23]. However, its specific role in AAK remains largely unexplored in the literature.

In summary, while PAX6 is a well-known transcription factor and a RA-inducible gene [24], its specific role in regulating the RA signaling pathway in AAK remains largely unexplored. In this study, our purpose was to investigate the expression of key regulators within the RA signaling pathway in response to varying concentrations of RA treatment, using an siRNA-mediated PAX6-knockdown aniridia LEC model, *in vitro*.

 

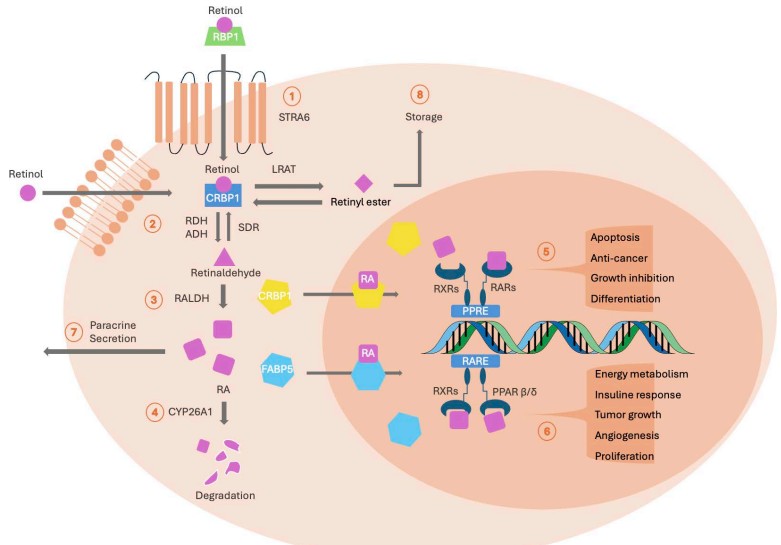

**Fig 1. RA signaling pathway regulates the dynamic balance of differentiation and proliferation in LECs.** Retinoic acid (RA) is the downstream metabolite of retinol, which is initially taken up into the cell by STRA6 (1). Once inside, retinol binds to cellular retinol-binding protein 1 (CRBP1), initiating a two-step oxidative process that converts it into functional RA (2, 3). Excess retinol can follow three alternative pathways: it can be stored in the cytoplasm as retinyl ester (8), secreted paracrinally to adjacent tissues (7), or degraded via CYP26A1 (4). As carrier proteins, cellular RA-binding protein 2 (CRABP2) and fatty acid-binding protein (FABP5) maintain a dynamic balance, both binding RA to transport it into the nucleus for downstream gene transcription [15,16]. RA transported by CRABP2 binds to the RXR-RAR heterodimer at the PPRE region, regulating apoptosis and suppressing cell growth (5) [15,16]. Alternatively, RA can activate the FABP5-bound PPARG axis, inducing transcription at the RARE region, which is essential for maintaining basic cell functions and promoting cell growth (6) [18,21,22]. STRA6: stimulated by retinoic acid 6, SDR: short-chain dehydrogenase/reductase, LRAT: lecithin retinol acyltransferase, PPRE: peroxisome proliferator-activated receptor response element, RARE: retinoic acid response element.

## 2 Materials and methods

The study was approved by the Ethical Committee of Saarland, Germany (Approval No.: 178/22). The recruitment period of the prospective study started from 07.11.2022 to 15.12.2024.

### 2.1 Cell culture

Limbal tissue from eight healthy donors (n = 8, age: 68.38 ± 21.68 years, range: 29–90 years, 88% male) (Table 1) was obtained from the LIONS Cornea Bank Saar-Lor-Lux, Trier/Westpfalz. A 1.5 mm biopsy of the corneal limbus was performed under light microscopy using the punching method. The collected tissue was incubated overnight at 37°C in keratinocyte growth medium (KGM3, PromoCell, Heidelberg, Germany) supplemented with collagenase (1 mg/ml) (Roche Diagnostic GmbH, Mannheim, Germany, No.10103578001) to facilitate tissue lysis. The following day, trypsin was used to detach the cells. The cell pellet was isolated via centrifugation (800 × g, 4–5 minutes) and re-seeded in 6-well plates with 2.5 ml of KGM3 per well. Cells were cultured at 37°C in a humidified incubator (95% relative humidity, 5% $CO_2$). Trypsinization was used to passage cells or to remove limbal fibroblasts, ensuring a > 95% corneal epithelial cell population. Only pure limbal epithelial cells (LECs) at passages 2–4 (P2–P4) with ≥80% confluence were used for subsequent siRNA treatments.

### 2.2 siRNA transfection and RA treatment

Each well of a 6-well plate was prepared with 2 ml of keratinocyte growth medium (KGM3) and treated with 500 μL Opti-MEM, 5 μL Lipofectamine, and 1 μL siRNA for 24 hours. Prior to transfection, Lipofectamine and siRNA were pre-incubated together in Opti-MEM at room temperature for 20 minutes before being added to the culture plate.

**Table 1. Demographic information of corneal donors.**

| Donor number | Gender | Age (years) |
|---|---|---|
| 1 | male | 72 |
| 2 | male | 43 |
| 3 | female | 72 |
| 4 | male | 29 |
| 5 | male | 70 |
| 6 | male | 88 |
| 7 | male | 90 |
| 8 | male | 83 |

For transfection, the scrambled siRNA (Negative Control #1 siRNA) was used in the PAX6 control group, while the PAX6-targeting siRNA (silencing PAX6 siRNA) was used in the PAX6 knockdown group. On the following day, the old KGM3 was replaced with 2.5 ml of fresh KGM3 for 24 hours to allow cell recovery from siRNA treatment. On day three, retinoic acid (RA) treatments were administered at concentrations of 0 µM, 1 µM, and 5 µM in both the siRNA knockdown and scrambled siRNA groups for an additional 48 hours of incubation. The RA concentrations used in this study are based on commonly reported doses in the literature that ensure sufficient activation of RA receptors [20,25]. Although all-trans retinoic acid (t-RA) was used for the treatment, we will refer to it as RA throughout the manuscript for clarity. At the end of the epithelial cultivation, supernatants were collected for ELISA analysis, and cells were harvested using 300 µL SKP (NORGEN RNA/DNA/Protein Purification Plus Kit, Biokat, Thorold, Canada) supplemented with 3 µL 2-Mercaptoethanol (Merck KGaA, Darmstadt, Germany) per well for further analysis.

## 2.3 RNA/ protein extraction and cDNA synthesis

RNA and protein were isolated using the RNA/DNA/Protein Purification Plus Kit according to the manufacturer's protocol. The RNA concentration was measured via UV/VIS spectrophotometry (Analytik Jena AG, Jena, Germany), and the obtained RNA was used for complementary DNA (cDNA) synthesis using the OneTaq® RT-PCR Kit (New England Biolabs Inc., Ipswich, USA), targeting a final concentration of 500 ng RNA. The protein concentration was determined using a Tecan Infinite F50 Absorbance Microplate Reader (Tecan Group AG, Männedorf, Switzerland) at 595 nm absorbance and standardized using Bradford's assay for subsequent Western blot (WB) analysis.

## 2.4 Quantitative PCR

For quantitative PCR (qPCR) analysis, a reaction mixture was prepared for each well of a 96-well plate, consisting of 3 µL RNase-free water, 1 µL primer (QIAGEN GmbH, Hilden, Germany) (Table 2), 5 µL AceQ SYBR qPCR Master Mix (Vazyme Biotech, Nanjing, China), and 1 µL cDNA. TATA-binding protein (TBP) was used as the reference gene. The amplification protocol was set to 40 cycles with the following conditions: 95°C for 10 seconds, 64°C for 10 seconds, and 72°C for 45 seconds. Cycle threshold (Ct) values were measured and normalized to the reference gene. Data analysis and fold-change calculations were performed using QuantStudio™ Design & Analysis software (ThermoFisher Scientific™, Dreieich, Germany), applying the $2^{\wedge}\Delta\Delta Ct$ method. Results were presented as the geometric mean ± geometric standard deviation (SD).

## 2.5 Western blot analysis

For Western blot (WB) analysis, 20 µg of protein from each sample was mixed with 5 µL Laemmli Sample Buffer (Bio-Rad Laboratories, Hercules, California, USA) and boiled at 95°C for 5 minutes before being loaded onto NuPAGE™ Bis-Tris

 

**Table 2. Primer pairs used for qPCR.**

| Referred as | Qiagen Cat. No | Amplicon size(bp) |
| --- | --- | --- |
| ADH7 | QT00000217 | 85 |
| ALDH1A1 | QT00013286 | 97 |
| CRABP2 | QT00063434 | 140 |
| CYP26A1 | QT00026817 | 150 |
| FABP5 | QT00225561 | 97 |
| PAX6 | QT00071169 | 113 |
| PPARG | QT00029841 | 113 |
| RARA | QT00095865 | 148 |
| RARB | QT00062741 | 118 |
| RBP1 | QT01850296 | 126 |
| RDH10 | QT00029176 | 107 |
| RXRA | QT00005726 | 134 |
| RXRB | QT00061117 | 92 |
| TBP | QT00000721 | 132 |
| VEGFa | QT01010184 | 273, 222, 204, 150* |

*VEGFA primer is a mix of NM_001025366 (transcript variant 1), NM_001171629 (transcript variant 7), NM_001171623(transcript variant 1), NM_001171625(transcript variant 3), NM_001025368(transcript variant 4), NM_001033756(transcript variant 7), NM_001025367(transcript variant 3), NM_001171624(transcript variant 2), NM_003376(transcript variant 2), NM_001171626(transcript variant 4), NM_001287044(transcript variant 10).

Precast 4–12% Bis-Tris gels (ThermoFisher Scientific™ GmbH, Dreieich, Germany). A 3 µL Precision Protein All Blue Standards (BIO-RAD) reagent was used as a molecular weight marker. The gel was run at a voltage between 80–120 V until the standard protein bands were well-separated and had reached the bottom of the gel. Following electrophoresis, proteins were transferred onto a nitrocellulose membrane using the Trans-Blot® Turbo™ Transfer System (Bio-Rad Laboratories, Hercules, California, USA). The membrane was then incubated with Invitrogen™ No-Stain™ Protein Labeling Reagent (ThermoFisher Scientific™ GmbH, Dreieich, Germany) for 10 minutes to visualize total protein levels for normalization. For target protein detection, the membrane was incubated overnight at 4°C with primary antibodies (Table 3), followed by immunolabeling with Western Lightning® Plus ECL Reagent (PerkinElmer, Inc., Waltham, USA). Images were captured using the iBright™ Imaging System (ThermoFisher Scientific™ GmbH, Dreieich, Germany). Before and after exposure, as well as before incubation with a new antibody, membranes were washed three times for 5 minutes using Western Froxx Wash Solution (10X) (neo-Froxx GmbH, Einhausen, Germany). Band intensity was normalized to total protein levels using total protein normalization (TPN) and corrected for local background. Data analysis was performed using the iBright™ Imaging System/iBright™ Analysis Software.

## 2.6 ELISA

In our study, only VEGFA protein expression was measured using ELISA due to its secretory nature. The assay was performed according to the manufacturer's protocol using the DuoSet® ELISA Kit (R&D Systems Europe, Ltd., Abingdon, UK). The protein concentration was determined using the Tecan Infinite F50 Absorbance Microplate Reader (Tecan Group AG, Männedorf, Switzerland) at an absorbance wavelength of 450 nm.

## 2.7 BrdU assay

LECs from eight healthy cornea donors were isolated and seeded in a 96-well plate at a density of 3,200 cells per well. Cells were cultured following the standard cell culture protocol (see 2.1), and siRNA treatment (see 2.2) was

**Table 3. Antibodies used for western blot analysis.**

| Antibody | Referred as | Catalog number | Dilution | Molecular weight |
|---|---|---|---|---|
| ADH Polyclonal antibody | ADH7 | #PA5–98484 | 1:500 | 40 kDa |
| ALDH1A1 (H-4): sc-374076 mouse monoclonal IgG | ALDH1A1 | #A2319 | 1:100 | 56 kDa |
| CRABP2 Monoclonal antibody | CRABP2 | #66468–1-Ig | 1:2500 | 14 kDa |
| CYP26A1 Polyclonal antibody | CYP26A1 | #28081–1-AP | 1:500 | 57 kDa |
| FABP5 Rabbit Polyclonal antibody | FABP5 | #12348–1-AP | 1:500 | 15 kDa |
| Anti-PAX6 rabbit Ab | PAX6 | #AB2237 | 1:1000 | 46, 48 kDa |
| PPAR Gamma Polyclonal antibody | PPARG | #16643–1-AP | 1:1000 | 60 kDa |
| RARA Polyclonal antibody | RARA | #10331–1-AP | 1:500 | 54 kDa |
| RARB Polyclonal antibody | RARB | #14013–1-AP | 1:1000 | 48 kDa |
| RDH10 Polyclonal antibody | RDH10 | #14644–1-AP | 1:1000 | 39 kDa |
| RXRA Polyclonal antibody | RXRA | #21218–1-AP | 1:2000 | 52 kDa |

initiated once the cells reached the desired confluence. After 12 hours, the medium was replaced, followed by 48 hours of RA incubation. The BrdU assay was performed according to the manufacturer's protocol (Merck KGaA, Darmstadt, Germany). For control wells, 100 µL KGM, 10 µL BrdU labeling solution, and 100 µL Anti-BrdU-POD were added. The results were measured using the Tecan Infinite F50 Absorbance Microplate Reader (Tecan Group AG, Männedorf, Switzerland).

## 2.8 Statistical analysis

All statistical analyses were performed using GraphPad Prism 10.0. A two-way ANOVA followed by the Dunnett post hoc test, was used to determine statistical significance. Results were considered significant for p-values $< 0.05$. Significance in expression levels was indicated in the figures, either in relation to RA treatment or PAX6 knockdown.

## 3 Results

### 3.1 PAX6 mRNA and protein levels following siRNA knockdown

PAX6 mRNA and protein levels were significantly reduced following PAX6 siRNA knockdown compared to the control siRNA group ($p < 0.001$ for both) (Fig 2).

### 3.3 mRNA and protein expression of key regulators in control and PAX6 knockdown LECs after retinoic acid treatment

First, qPCR was conducted to identify markers that exhibited significant changes in response to either PAX6 activity or RA treatment. For markers with changes at qPCR level, Western blot (WB) or ELISA was conducted to further assess changes in protein expression levels (Figs 3–6). Markers with protein expression levels too low for detection via WB analysis were validated using a positive control, which is provided in the supplementary data (S1 Fig).

Regarding enzymes related to the RA signaling pathway, *ADH7* and *ALDH1A1* mRNA expression levels (p = 0.018, p = 0.015) and protein levels (p = 0.003, p < 0.001) were significantly reduced in PAX6 knockdown LECs compared to control siRNA-treated LECs. Additionally, *ALDH1A1* mRNA levels were significantly decreased following 5 µM RA treatment in the control group (p = 0.038). Furthermore, *CYP26A1* mRNA expression was significantly upregulated

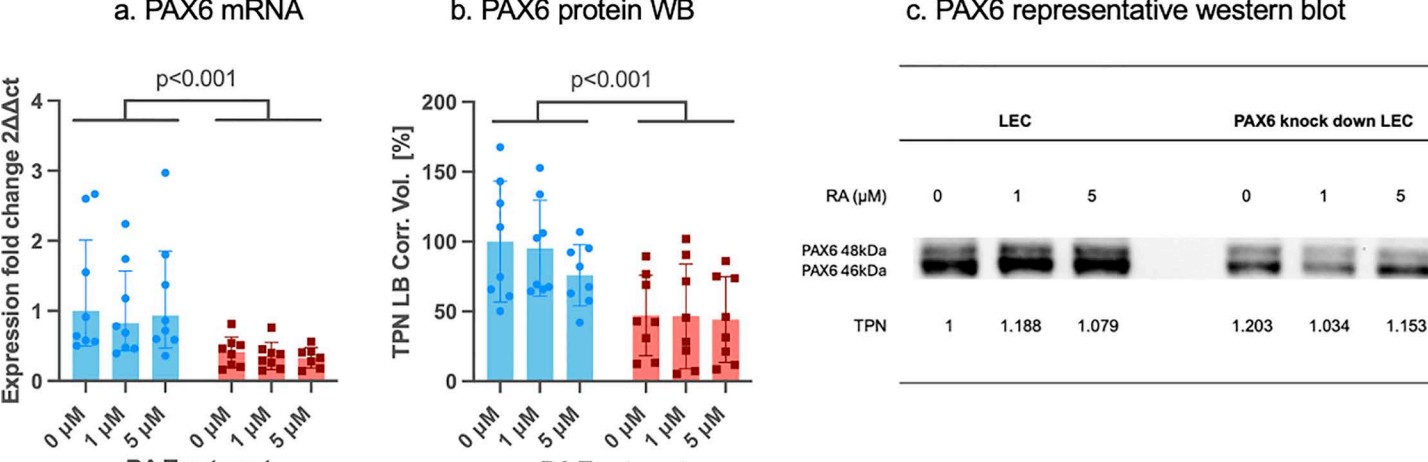

**Fig 2. PAX6 mRNA and protein levels (b,c) in control siRNA (blue bars) and in PAX6 knockdown (red bars) limbal epithelial cells (LECs) without and after 1µM or 5µM RA incubation for 48 hours.** mRNA values are shown as geometric mean±geometric standard deviation (SD). Total protein staining was used for protein normalization of each lane. The total protein normalization factor TPN is indicated below each lane. Band intensity from WB is shown with mean±SD. Two-way ANOVA, followed by Dunnett's test was used. Significant p values (<0.05) are highlighted. PAX6 mRNA (a) and protein levels (b) were significantly lower after PAX6 siRNA knockdown than using control siRNA (p<0.001 for both). Nevertheless, there was no significant difference between any other groups without or using different RA concentrations (p≥0.611).

after 1 µM and 5 µM RA treatment in both the control group (p<0.001, p<0.001) and PAX6 knockdown LECs (p=0.001, p=0.002) (Fig 3).

FABP5 mRNA and protein expression levels were significantly reduced in PAX6 knockdown LECs compared to control siRNA-treated LECs (p<0.001, p<0.001). Following 5 µM RA treatment, *CRABP2* mRNA expression was significantly downregulated in the PAX6 knockdown group (p=0.02), while CRABP2 protein levels were decreased in both the control (p=0.003) and PAX6 knockdown groups (p=0.02) (Fig 4).

Regarding the receptor's protein, PPARG2, its mRNA expression was significantly upregulated, while PPARG2 and RARB protein levels were significantly downregulated in PAX6 knockdown LECs compared to control siRNA-treated LECs (p=0.012, p=0.007, p<0.001). Following 5 µM RA treatment in the control group, *RARB* mRNA expression and RXRA protein levels were significantly downregulated (p=0.007, p=0.007). Additionally, RARB protein levels were reduced after both 1 µM and 5 µM RA treatment in the control group (p=0.02, p=0.004). In the PAX6 knockdown group, *RARA* and *RARB* mRNA expression were significantly downregulated following 5 µM RA treatment (p=0.023, p<0.001) (Fig 5).

Following 5 µM RA treatment, VEGFA mRNA expression levels were significantly upregulated in the control group (p=0.041) (Fig 6).

### 3.4 BrdU assay in control and PAX6 knockdown LECs after retinoic acid treatment

The BrdU assay revealed a significantly lower proliferation rate in the PAX6 siRNA knockdown group compared to the control siRNA group (p<0.001). Additionally, 5 µM RA treatment led to a significant reduction in proliferation rate in the control group (p=0.025) (Fig 7).

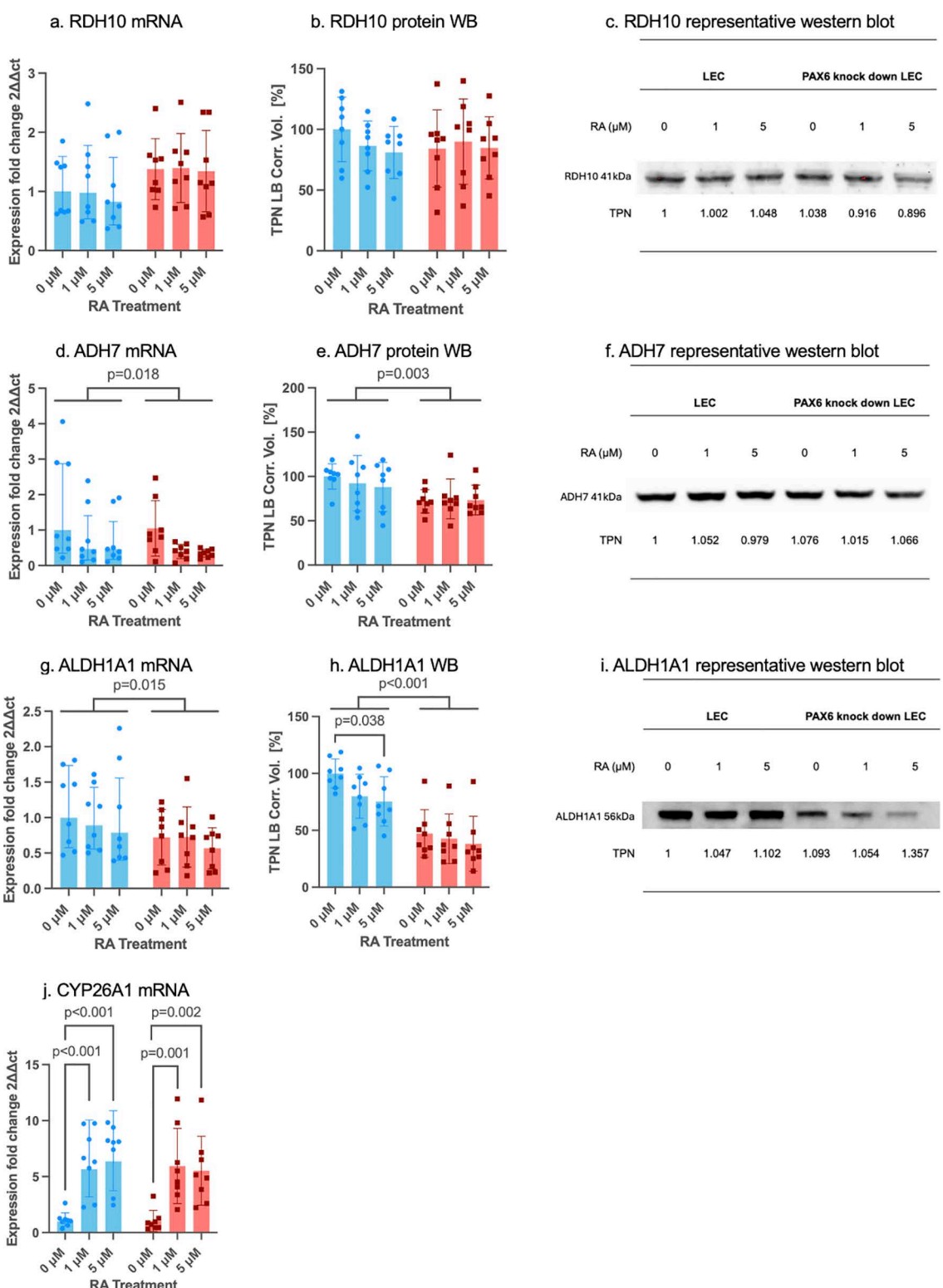

**Fig 3. mRNA and protein expression of enzymes in the retinoic acid signaling pathway in control siRNA (blue bars) and in PAX6 knock-down (red bars) limbal epithelial cells (LECs) without and after 1μM or 5μM RA incubation for 48 hours.** *RDH10* (a), *ADH7* (d), *ALDH1A1* (g), *CYP26A1* (j) mRNA and RDH10 (b), ADH7 (e), ALDH1A1 (h) protein levels (Western blot) are shown. A representative western blot of RDH10 (c), ADH7 (f), and ALDH1A1 (g) is displayed. mRNA results are shown as geometric mean ± geometric standard deviation (SD), and protein results are shown as

mean ± standard deviation, with significant differences indicated. Two-way ANOVA, followed by Dunnett`s test has been used. Including all analyzed groups, ADH7 and ALDH1A1 mRNA expression levels (p = 0.018, p = 0.015) and protein levels (p = 0.003, p < 0.001) were significantly downregulated in PAX6 knockdown LECs, compared to control siRNA treated LECs. ALDH1A1 mRNA levels were significantly downregulated after 5 µM RA treatment in the control group (p = 0.038). In addition, after 1 µM and 5 µM RA treatment, CYP26A1 mRNA expression was upregulated in control (p < 0.001, p < 0.001) and PAX6 knockdown LECs (p = 0.001, p = 0.002). Nevertheless, there was no significant difference between any other groups without or using different RA concentrations (p ≥ 0.077).

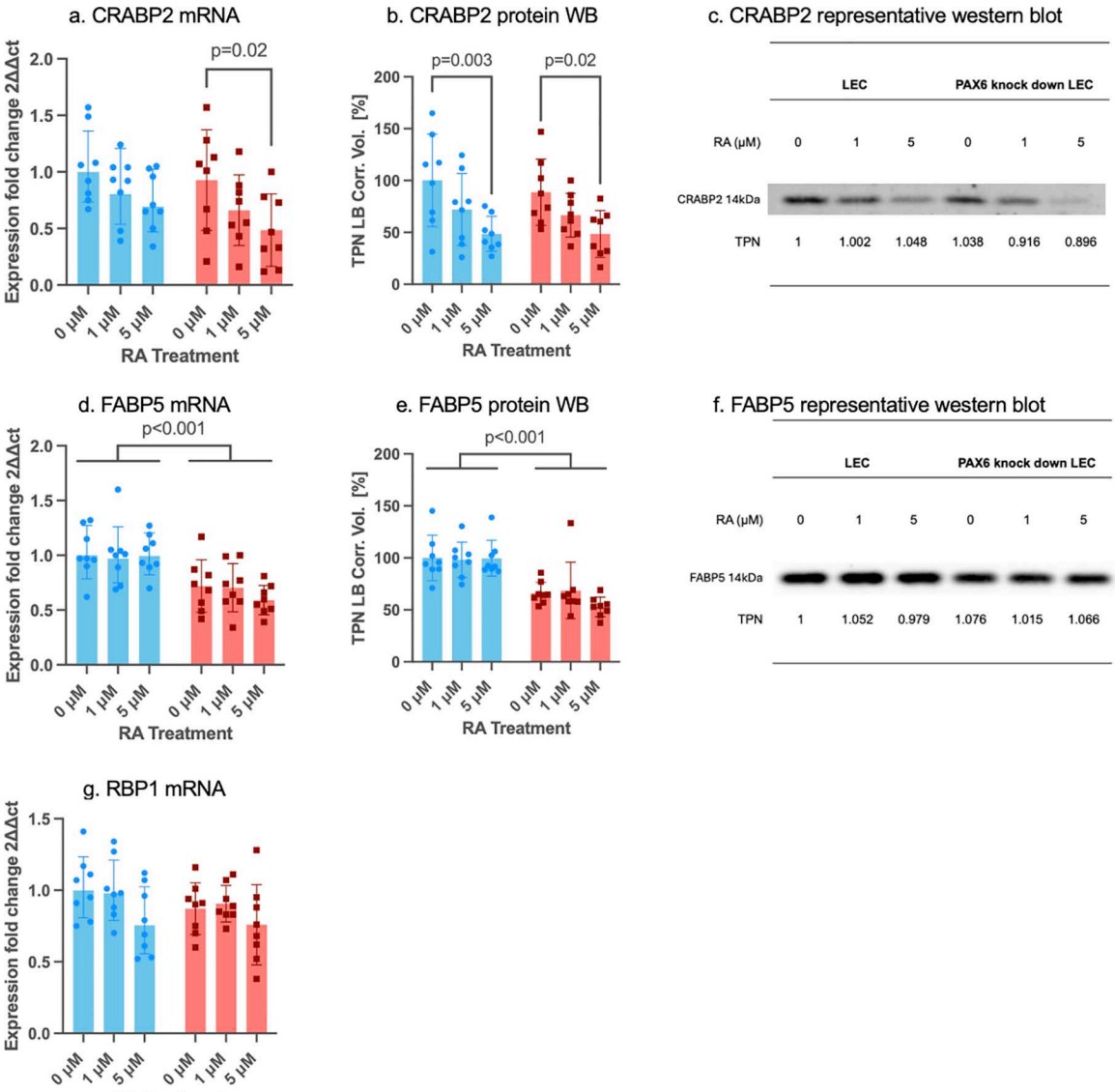

**Fig 4. mRNA and protein expression of binding proteins in control siRNA (blue bars) and in PAX6 knockdown (red bars) limbal epithelial cells (LECs) without and after 1µM or 5µM RA incubation for 48 hours.** CRABP2 (a), FABP5 (d), RBP1 (g) mRNA and CRABP2 (b), FABP5 (e) protein levels (Western blot) are shown. A representative western blot of CRABP2 (c) and FABP5 (f) is displayed. mRNA results are shown as geometric mean ± geometric standard deviation (SD), and protein results are shown as mean±standard deviation, with significant differences indicated. Two-way ANOVA, followed by Dunnett`s test has been used. Including all analyzed groups, mRNA and protein expression levels of FABP5 were significantly lower in PAX6 knockdown LECs, than in control siRNA-treated LECs (p < 0.001, p < 0.001). Following 5 µM RA treatment, CRABP2 mRNA level in the PAX6 knockdown group (p = 0.02) and CRABP2 protein level in both the control and the PAX6 knockdown groups was downregualted (p = 0.003, p = 0.02). Nevertheless, there was no significant difference between any other groups without or using different RA concentrations (p ≥ 0.067).

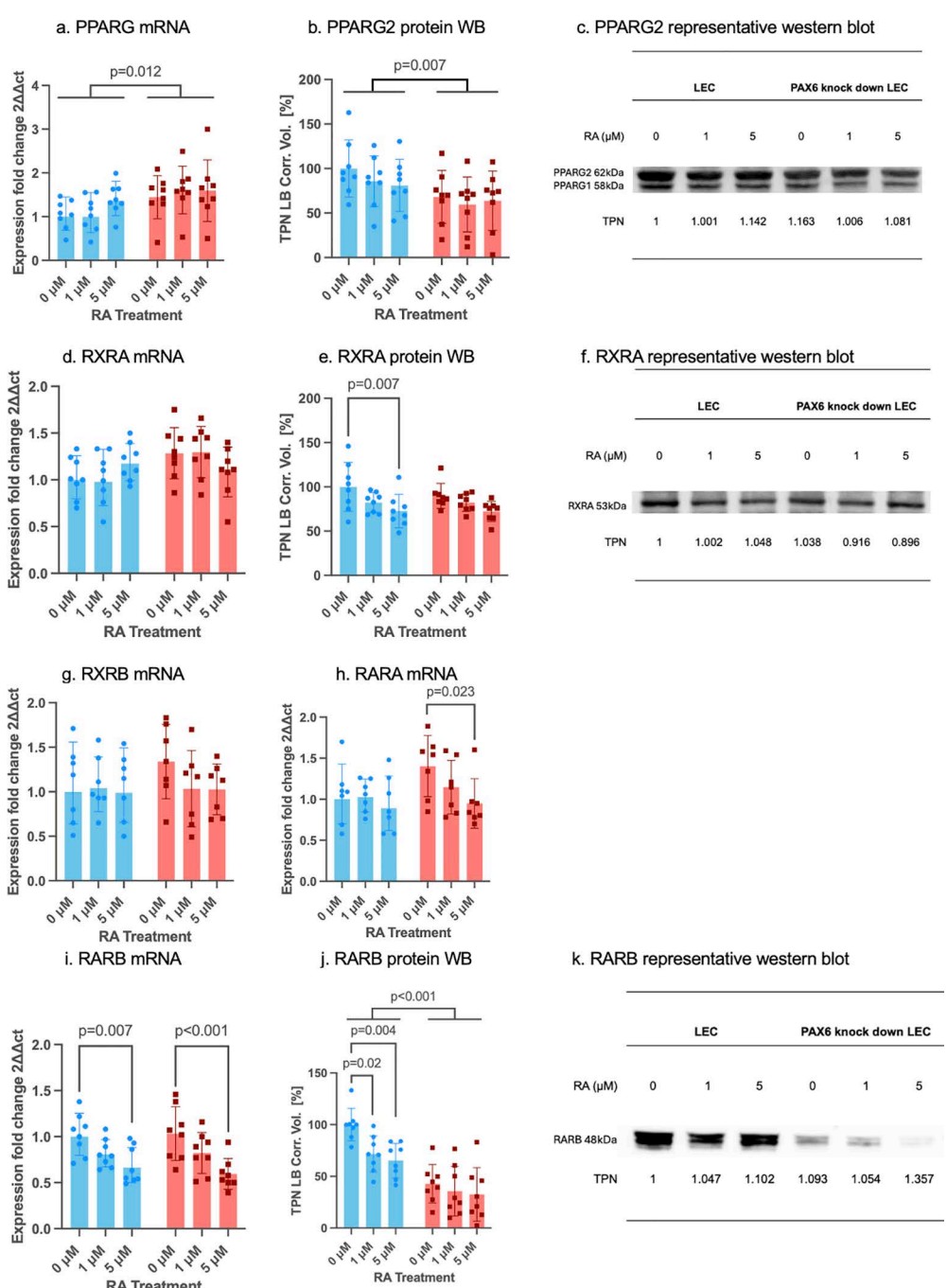

**Fig 5. mRNA and protein expression of receptors in RA signaling pathway in control siRNA (blue bars) and in PAX6 knockdown (red bars) limbal epithelial cells (LECs) without and after 1µM or 5µM RA incubation for 48 hours.** *PPARG* (a), *RXRA* (d), *RXRB* (g), *RARA* (h), *RARB* (i) mRNA and PPARG2 (b), RXRA (e) and RARB (j) protein levels (Western blot) are shown. A representative western blot of PPARG2 (c), RXRA (f), and RARB (k) is displayed. mRNA results are shown as geometric mean ± geometric standard deviation (SD), and protein results are shown as mean ± standard deviation, with significant differences indicated. Two-way ANOVA, followed by Dunnett`s test has been used. PPARG2 mRNA level was significantly upregulated and PPARG2 and RARB protein levels were significantly downregulated in PAX6 knockdown LECs, compared to control siRNA-treated LECs (p = 0.012, p = 0.007, p < 0.001). Following 5µM RA treatment, in the control group, *RARB* mRNA expression and RXRA protein expression were downregulated (p = 0.007, p = 0.007). After 1µM and 5µM RA treatment, RARB protein level was downregulated in the control group (p = 0.02, p = 0.004). After 5µM RA treatment, in the PAX6 knockdown group, *RARA* and *RARB* mRNA levels were significantly downregulated (p = 0.023, p < 0.001). Nevertheless, there was no significant difference between any other groups without or using different RA concentrations (p ≥ 0.054).

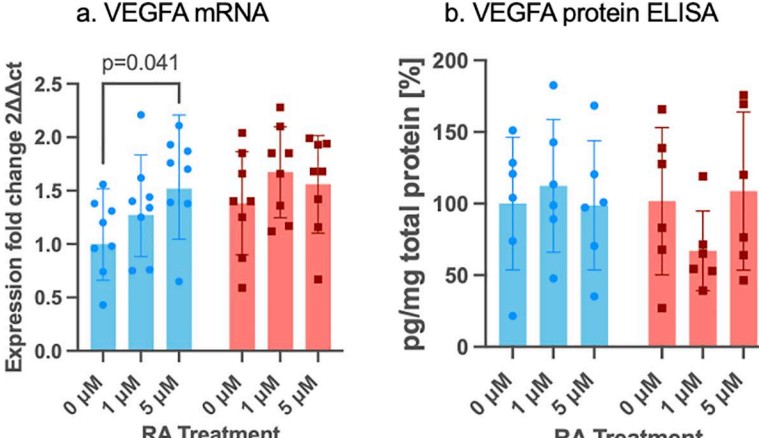

**Fig 6. mRNA and protein expression of VEGFA in control siRNA (blue bars) and in PAX6 knockdown (red bars) limbal epithelial cells (LECs) without and after 1µM or 5µM RA incubation for 48 hours.** *VEGFA* mRNA (a) and protein (Western blot) (b) expression are shown. mRNA results are shown as geometric mean ± geometric standard deviation (SD), and protein results are shown as mean ± standard deviation, with significant differences indicated. Two-way ANOVA, followed by Dunnett`s test has been used. Following 5µM RA treatment, VEGFA mRNA expression levels were significantly upregulated in controls (p = 0.041). Nevertheless, there was no significant difference between any other groups without or using different RA concentrations (p ≥ 0.127).

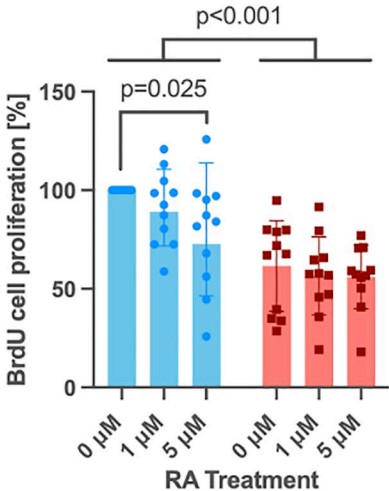

**Fig 7. BrdU assay in control siRNA (blue bars) and in PAX6 knockdown (red bars) limbal epithelial cells (LECs) without and after 1µM or 5µM RA incubation for 48 hours.** Two-way ANOVA, followed by Dunnett's test was used. Significant p values (<0.05) are highlighted. BrdU assay shows a significantly lower proliferation rate in the PAX6 siRNA knockdown group than in control siRNA group (p < 0.001). 5µM RA treatment significantly decreased proliferation rate in controls (p = 0.025). Nevertheless, there was no significant difference between any other groups without or using different retinol concentrations (p ≥ 0.455).

## 4 Discussion

Under normal physiological conditions, retinol circulates in the bloodstream and is taken up by cells, where it is metabolized into RA—the biologically active compound that initiates the RA signaling pathway. As a downstream metabolic product of retinol, RA elicits similar effects on LECs as retinol and its derivatives [26,27]. In most studies, RA is preferred over

retinol for treatment when investigating changes in the RA signaling pathway, as it is approximately 160 times more potent as an inducer [26]. Some research suggests that alterations in the expression levels of key markers may not be detectable within 24 hours of retinol treatment, but become evident after 48 hours [20]. This delay could be attributed to the time required for retinol conversion into RA in LECs, as well as the weaker cellular response elicited by retinol compared to RA.

The findings of this study demonstrate a downregulation of ADH7 and ALDH1A1 at both the mRNA and protein levels in siRNA-mediated PAX6 knockdown LECs, independent of RA concentration. These results are consistent with a previous study involving retinol treatment (Table 4) [28]. The downregulation of ADH7 and ALDH1A1 mRNA expression levels were not only observed in the primary aniridia cells [11], but also in the siRNA PAX6 knockdown LECs [11] (Table 5). This suggests that their downregulation is strongly linked to PAX6 activity rather than being a secondary effect of aniridia disease states. These findings further indicate that LECs with PAX6 haploinsufficiency may exhibit lower RA levels [20]. A localized reduction in RA could affect the expression of downstream regulators sensitive to RA concentration, potentially leading to altered cellular functions.

The present study indicates that CRABP2 expression is highly dependent on RA concentration rather than PAX6 activity (Table 5). However, previous studies have reported CRABP2 downregulation in association with PAX6 activity in both aniridia LECs and siRNA-mediated PAX6 knockdown LECs (Tables 4 and 5). Regarding the downregulation of retinol metabolic enzymes as previously discussed, the lower CRABP2 expression observed in PAX6-knockdown LECs may be more closely linked to reduced RA levels rather than the absence or deficiency of PAX6. Additionally, CRABP2 downregulation has been described as a response to increased retinoid concentration in healthy LECs [20]. Our results suggest that CRABP2 expression and its associated receptors (RARA/RARB) are tightly regulated in LECs in response to RA concentration. Although CRABP2 does not appear to be directly regulated by PAX6, a reduced CRABP2-associated differentiation pathway may still be present in PAX6 haploinsufficient LECs, likely due to defective RA signaling pathway.

**Table 4. mRNA and protein expression changes in siRNA PAX6 knockdown LECs following retinol and retinoic acid (RA) treatment. Some mRNA and protein level changes were related to the PAX6 knockdown, some others to the treatment, as described in the table.**

| Category | Markers | Retinol [28] | | RA (present study) | |
|---|---|---|---|---|---|
| | | mRNA | protein | mRNA | protein |
| RA metabolic pathway | RBP1 | n.s. | – | n.s. | – |
| | ADH7 | ↓ knockdown | n.s. | ↓ knockdown | ↓knockdown |
| | ALDH1A1 | ↓ knockdown | ↓knockdown | ↓ knockdown | ↓ treatment ↓ knockdown |
| | CYP26A1 | ↑ treatment | – | ↑ treatment | – |
| CRABP2 pathway | CRABP2 | ↓ knockdown | ↓ knockdown | ↓ treatment | ↓ treatment |
| | RARA | n.s. | – | ↓ treatment | – |
| | RARB | n.s. | – | ↓ treatment | ↓ treatment ↓ knockdown |
| | RXRA | n.s. | – | n.s. | ↓ treatment |
| | RXRB | n.s. | – | n.s. | – |
| FABP5-PPARG axis | FABP5 | ↓ knockdown | n.s. | ↓ knockdown | ↓ knockdown |
| | PPARG | n.s. | ↓ knockdown | ↑ knockdown | ↓ knockdown |

upregulation ↑ (orange background color), downregulation ↓ (blue background color), no significant change n.s., not performed.

Retinol-binding protein 1 (RBP1), alcohol dehydrogenase 7 (ADH7), aldehyde dehydrogenase 1 family member A1 (ALDH1A1), cytochrome P450 family 26 subfamily A member 1 (CYP26A1), cellular retinoic acid-binding protein 2 (CRABP2), retinoic acid receptor alpha (RARA), retinoic acid receptor beta (RARB), retinoid X receptor alpha (RXRA), retinoid X receptor beta (RXRB), fatty acid-binding protein 5 (FABP5), peroxisome proliferator-activated receptor gamma (PPARG), retinoic acid (RA).

**Table 5. mRNA expression level changes in conjunctival and corneal epithelial cells following retinol, retinoic acid and RAR antagonist treatment.**

| Reference | [9,10] | [11] | [11] | [20] | [20] | [28] | [present study] |
|---|---|---|---|---|---|---|---|
| Markers | aniridia (PAX6$^{+/-}$) conjunctiva | aniridia (PAX6$^{+/-}$) LECs | siRNA PAX6$^{+/-}$LECs | LEC+retinol/ RA | LEC+RAR antagonist | siRNA PAX6$^{+/-}$ LECs+ retinol | siRNA PAX6$^{+/-}$ LECs+RA |
| RBP1 | ↑ | n.s. | ↓ | ↑/ n.s. | n.s. | n.s. | n.s. |
| ADH7 | ↓ | ↓ | ↓ | n.s./ n.s. | ↑ | ↓knockdown | ↓ knockdown |
| RDH10 | ↑ | n.s. | ↓ | n.s./↑ | ↑ | ↑ knockdown | n.s. |
| ALDH1A1 | – | ↓ | ↓ | n.s./ n.s. | n.s. | ↓ knockdown | ↓ knockdown |
| CYP26A1 | – | – | – | – | – | ↑treatment | ↑treatment |
| CRABP2 | ↑ | ↓ | ↓ | ↓/↓ | ↑ | ↓ knockdown | ↓treatment |
| RARA | – | – | – | – | – | n.s | ↓treatment |
| RARB | – | – | – | – | – | n.s | ↓treatment |
| RXRA | – | – | – | – | – | n.s | n.s |
| RXRB | – | – | – | – | – | n.s | n.s |
| FABP5 | ↓ | ↓ | ↓ | n.s./ n.s. | ↑ | ↓ knockdown | ↓ knockdown |
| PPARG | ↑ | ↑ | ↓ | ↓/ n.s. | ↑ | n.s. | ↑ knockdown |

upregulation ↑ (orange background color), downregulation ↓ (blue background color), no significant change n.s., not performed.

Paired box 6 (PAX6), retinol-binding protein 1 (RBP1), alcohol dehydrogenase 7 (ADH7), retinol dehydrogenase 10 (RDH10), aldehyde dehydrogenase 1 family member A1 (ALDH1A1), cytochrome P450 family 26 subfamily A member 1 (CYP26A1), cellular retinoic acid-binding protein 2 (CRABP2), retinoic acid receptor alpha (RARA), retinoic acid receptor beta (RARB), retinoid X receptor alpha (RXRA), retinoid X receptor beta (RXRB), fatty acid-binding protein 5 (FABP5), peroxisome proliferator-activated receptor gamma (PPARG), retinoic acid (RA), limbal epethelail cells (LECs).

In the present study, a downregulation pattern was observed in multiple RAR and RXR receptors in response to increased RA concentration, with RARs exhibiting more pronounced changes compared to RXRs. However, previous studies found no significant changes in RAR and RXR mRNA expression levels in response to retinol treatment (Table 4) [28]. This discrepancy may be attributed to the higher potency of RA compared to retinol, as discussed previously. Additionally, RARs have a higher binding affinity and greater sensitivity to all-trans RA (t-RA), whereas RXRs exhibit lower affinity for t-RA and primarily respond to 9-cis-RA [29]. Since t-RA was used in this experiment, it primarily binds to RARs, regulating RXR-RAR heterodimers. However, a fraction of t-RA may undergo isomerization to form 9-cis-RA, which can bind to both RXRs and RARs [30]. Notably, 9-cis-RA is an unavoidable contaminant in t-RA solutions, potentially leading to over-activation of RXR-RAR heterodimers [29]. In summary, RXRs can be activated either by direct binding to cis-RA or indirectly through RXR-RAR heterodimer activation, contributing to the less pronounced changes observed in RXRs in response to RA treatment in our study.

The activation of the FABP5-bound PPARG axis has been linked to cell growth [31]. In our study, we observed a downregulation of FABP5 and PPARG protein expression, accompanied by a decreased proliferation rate in the BrdU assay in PAX6-knockdown LECs. These findings are consistent with previous research by Latta et al., which also reported FABP5 mRNA downregulation in both aniridia LECs and PAX6-knockdown LECs [11]. This suggests that LECs with PAX6 haplo-insufficiency may exhibit reduced proliferative potential.

Additionally, the BrdU assay revealed that cell proliferation was reduced not only due to PAX6 knockdown but also in response to increased RA concentration in control LECs. The effect of RA concentration on cell proliferation and

differentiation is highly context-dependent [32–34]. Most studies suggest that low to moderate RA concentrations promote cell proliferation [35], whereas higher RA concentrations induce cell cycle arrest and potentially apoptosis, serving as a protective mechanism against uncontrolled cell division [36,37]. Interestingly, a previous study using retinol instead of RA reported an increased tendency in cell proliferation following 5 µM retinol treatment [28]. This discrepancy may stem from the fact that retinol must be metabolized into RA, leading to a more gradual and reduced effect compared to the direct application of 5 µM RA. Consequently, cells treated with retinol may favor proliferation over differentiation.

Additionally, higher RA concentrations may inhibit cell proliferation through RAR activation, a mechanism independent of FABP5 expression [38]. This is supported by our observation that FABP5 mRNA and protein expression levels remained unchanged in response to RA treatment. Since RA preferentially binds to RXR-RAR heterodimers) [29], an increase in RA concentration may promote RXR binding to RARs instead of PPARG. As a result, RXR protein levels may be depleted in the process [39]. In other words, FABP5 expression alone is insufficient to activate the RXR-PPARG pathway for cell proliferation without an adequate supply of RXRs.This hypothesis is further supported by findings from Latta et al., who reported that FABP5 mRNA expression was upregulated in LECs treated with RAR antagonists [20]. Based on our findings, we propose that FABP5 is not an RA-inducible gene, and its expression alone cannot reliably predict the activation of the RXR-PPARG pathway or cell proliferation rates.

The pathogenesis of AAK in congenital aniridia has been proposed to result from corneal limbal stem cell deficiency (LSCD) [40]. This theory is supported by histopathological changes, such as limbal vascularization and the presence of conjunctiva-derived goblet cells in the corneal epithelium, which are considered early indicators of AAK [41]. However, direct evidence of limbal stem cell dysfunction remains inconclusive, as a definitive limbal stem cell marker has not yet been identified [42]. Recent clinical observations and animal model studies suggest that AAK may be more closely linked to alterations in the limbal stem cell niche rather than to an inherent stem cell deficiency [40]. The underlying mechanism included: (1) Impaired corneal healing response, characterized by defective centripetal migration of epithelial cells and (2) Abnormal epithelial differentiation, leading to increased fragility and susceptibility to shearing forces, as frequently observed in AAK patients [42]. Over time, these cumulative corneal surface injuries become more pronounced with aging, ultimately contributing to progressive disruption of the limbal stem cell niche [43]. This is supported by K. Ramaesh et al., who reported asymmetrical corneal changes in aniridia patients, with more severe AAK progression in eyes that had undergone cataract surgery via corneal incision [42]. Additionally, in two out of twenty cases, AAK progression worsened following traumatic corneal abrasions, further highlighting a potentially abnormal healing process in the aniridic cornea [42]. This is consistent with studies suggesting that PAX6 continues to play a regulatory role in adulthood [44], and its expression increases during corneal epithelial wound healing [42]. In our study, the proliferation rate in siRNA PAX6-knockdown LECs was significantly lower and less responsive to RA treatment, indicating a potential alteration in the regulation of cell proliferation. These findings further support the hypothesis that PAX6 plays a critical role in maintaining corneal epithelial homeostasis and wound healing, and its deficiency may contribute to AAK progression.

Our results suggest that the functions of RDH10 and CYP26A1 remain preserved in LECs despite PAX6 knockdown. The increased mRNA expression of CYP26A1 in response to elevated RA concentrations in both the PAX6 control and knockdown groups indicates that LECs actively eliminate excess RA when its concentration rises. However, since CYP26A1 activity remains unaffected by PAX6 knockdown, this suggests that RA metabolism operates independently of PAX6 activity. These findings imply that siRNA-PAX6 knockdown LECs possibly retain their ability to degrade excessive RA products, maintaining a functional RA degradation pathway despite PAX6 deficiency.

In this study, we focused on changes in the expression levels of key regulators in response to PAX6 knockdown and RA treatment, while also providing a comprehensive comparison with existing literature. However, several limitations of our siRNA PAX6 knockdown LEC model must be acknowledged.

First, the influence of the aniridia disease status and potential crosstalk between different regulatory pathways were not fully considered. It has been reported that the expression of RBP1, RDH10, and CRABP2 can be influenced both by

altered retinoid levels and by the activation of RARs and RXRs [20]. Additionally, ADH7 expression may be suppressed by RAR and RXR activation [20], and is functionally linked to medium-chain fatty acid metabolism, which in turn is associated with FABP5 expression [18]. Studies have also highlighted the dual role of FABP5 in both cell proliferation and differentiation, further complicating its regulatory functions [45,46].

Furthermore, there remains a possibility that t-RA influences gene expression indirectly through post-translational regulation of PAX6 [20,24]. Although RA-induced PAX6 expression is well documented during early developmental stages, it appears to be highly context-dependent. Our results suggest that in adult LECs, PAX6 is not directly upregulated by RA under the conditions tested. To minimize potential off-target effects of siRNA treatment, we used a commercially validated PAX6-targeting siRNA and compared it to a scrambled siRNA negative control.

Additionally, the LEC population used in this study was derived from punch biopsies of the corneal limbus region, with limbal fibroblast cells (LFCs) removed through trypsinization to specifically assess the role of LECs in RA treatment. However, LFC-derived growth factors have been reported to play a positive role in keratinocyte differentiation and contribute to the maintenance of the limbal stem cell niche [47,48]. Changes in the limbal stem cell niche are widely proposed as a major factor in limbal stem cell deficiency in AAK [42]. By eliminating LFCs, our siRNA PAX6 knockdown model does not account for cell-cell interactions between LECs and LFCs, nor the cell-matrix interactions that are crucial in the limbal stem cell niche.

Therefore, to gain a more comprehensive understanding of RA signaling in AAK, future studies should extend beyond the analysis of key regulator expression levels to avoid oversimplifying the complexity of the disease. Additional approaches, such as functional assays, metabolomic profiling, and co-culture models, may offer deeper insights into the interplay between RA metabolism, PAX6 activity, and the limbal microenvironment.

## 5 Conclusions

Our results demonstrate downregulation of the FABP5-bound PPARG axis and retinol metabolic enzymes, including ADH7 and ALDH1A1, in PAX6-knockdown LECs compared to controls. In contrast, the expression levels of CRABP2 and multiple receptor proteins (RXRs and RARs) were downregulated in response to increased RA concentration but remained independent of PAX6 activity. Additionally, RDH10 and CYP26A1 expression levels were unaffected by PAX6 haploinsufficiency, suggesting that retinol metabolism may remain functionally preserved in PAX6 haploinsufficient LECs.

Furthermore, siRNA-mediated PAX6 knockdown LECs exhibited a lower proliferation rate and reduced responsiveness to RA concentration, indicating a potential impairment in cell proliferation regulation.

This study highlights altered expression levels of key regulators in the RA signaling pathway following PAX6 knockdown and RA treatment, along with a reduced proliferation rate in PAX6-knockdown LECs. Future studies incorporating additional functional analyses are necessary to gain a deeper understanding of AAK pathophysiology and to explore the therapeutic potential of RA-based treatments.

## Supporting information

**S1 Fig. CYP26A1 and RARA Western blot analysis.** CYP26A1 (a) and RARA (b) western blot show low protein expression levels in limbal epithelial cells (LECs), compared to positive control Hep G2 and MCF 7 for CYP26A1 and MCF7 for RARA.
(DOCX)

**S1 File. Raw_images.**
(PDF)

## Author contributions

**Conceptualization:** Tanja Stachon, Fabian N. Fries, Nóra Szentmáry.

**Data curation:** Shao-Lun Hsu.

**Formal analysis:** Shao-Lun Hsu.

**Funding acquisition:** Tanja Stachon, Nóra Szentmáry.

**Investigation:** Shao-Lun Hsu.

**Methodology:** Shao-Lun Hsu.

**Project administration:** Nóra Szentmáry.

**Resources:** Shao-Lun Hsu, Fabian N. Fries, Swarnali Kundu.

**Software:** Shao-Lun Hsu.

**Supervision:** Tanja Stachon, Nóra Szentmáry.

**Validation:** Shao-Lun Hsu.

**Visualization:** Shao-Lun Hsu.

**Writing – original draft:** Shao-Lun Hsu.

**Writing – review & editing:** Tanja Stachon, Fabian N. Fries, Zhen Li, Shuailin Li, Shanhe Liu, Berthold Seitz, Swarnali Kundu, Maryam Amini, Shweta Suiwal, Nóra Szentmáry.

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
