## [Decision Letter · Decision Letter 0]

PONE-D-25-11947Effect of Retinoic Acid Treatment on the Retinoic Acid Signaling Pathway in an siRNA-Based Aniridia Limbal Epithelial Cell Model, In VitroPLOS ONE

Dear Dr. Hsu,

Thank you for submitting your manuscript to PLOS ONE. After careful consideration, we feel that it has merit but does not fully meet PLOS ONE’s publication criteria as it currently stands. Therefore, we invite you to submit a revised version of the manuscript that addresses the points raised during the review process.

The paper has been reviewed and some concerns voiced.

1. As the study uses two RA concentrations, please provide a rationale for selecting these specific doses and whether they compare to physiological levels of RA in limbal epithelial cells. Were dose-response experiments conducted to determine these concentrations?

2. It is unclear what happened to protein expression in control siRNA transfected cells. At least some westerns should cover this gap. The manuscript does not discuss potential off-target effects of siRNA.

3. Please elaborate on statistical analysis as suggested.

4. Was there any effect of siRNA on cell migration (from your or literature data)?

5. Please clarify the issue about RA treatment influence on PAX6 as suggested.

6. In the title, please consider adding the word “human”.

We look forward to receiving your revised manuscript.

Kind regards,

Alexander V Ljubimov, Ph.D.

Academic Editor

PLOS ONE

Journal Requirements:

Reviewers' comments:

Reviewer's Responses to Questions

**Comments to the Author**

1. Is the manuscript technically sound, and do the data support the conclusions?

Reviewer #1: Yes

2. Has the statistical analysis been performed appropriately and rigorously? 

Reviewer #1: N/A

3. Have the authors made all data underlying the findings in their manuscript fully available?

Reviewer #1: No

4. Is the manuscript presented in an intelligible fashion and written in standard English?

Reviewer #1: Yes

5. Review Comments to the Author

Reviewer #1: This study investigates the interplay between retinoic acid (RA) signaling and PAX6 deficiency in limbal epithelial cells (LECs), a model relevant to aniridia-associated keratopathy (AAK). By knocking down PAX6 in primary human LECs and treating with RA, the research reveals significant alterations in gene/protein expression of RA metabolic enzymes (e.g., ALDH1A1, CYP26A1), binding proteins (CRABP2), receptors (RXRA, RARA/B), and downstream targets (VEGF-A, PPARG). Notably, PAX6 knockdown disrupted RA-responsive pathways, while exogenous RA dose-dependently modulated these effectors, particularly suppressing cell proliferation in controls. These findings highlight RA signaling as a potential therapeutic target in AAK pathogenesis by elucidating its dysregulation in PAX6-deficient LECs. However, some concerns need to be addressed properly to strengthen the rigour of these findings.

Major

1. The study uses 1µM and 5µM RA concentrations, but the rationale for selecting these specific doses is not fully explained, the author should justify the RA concentrations used in this study. Since RA is highly potent, were dose-response studies or pilot experiments conducted to determine these concentrations? Additionally, how do these doses compare to physiological levels of RA in limbal epithelial cells?

2. The manuscript does not discuss potential off-target effects of siRNA. Can the authors confirm that the observed changes in RA pathway genes are directly due to PAX6 knockdown rather than siRNA transfection artifacts?

Minor

3. The manuscript states that statistical analysis was performed using two-way ANOVA followed by Dunnett’s test, but p-values for multiple comparisons should be clarified. Were adjustments for multiple testing (e.g., Bonferroni correction) applied?

4. The term “retinoic acid” is sometimes abbreviated as RA, but in other places, t-RA (all-trans retinoic acid) is used. Please ensure consistency.

5. The introduction describes PAX6 as an RA-inducible gene (line 158), but later, the study suggests PAX6 levels remain unchanged upon RA treatment (Fig.2). This potential contradiction should be clarified.

6. PLOS authors have the option to publish the peer review history of their article (what does this mean? ). If published, this will include your full peer review and any attached files.

**Do you want your identity to be public for this peer review?** For information about this choice, including consent withdrawal, please see our Privacy Policy .

Reviewer #1: No

---

## [Author Response · Author response to Decision Letter 1]

23 Apr 2025

[Comments from Academic Editor]

Thank you for submitting your manuscript to PLOS ONE. After careful consideration, we feel that it has merit but does not fully meet PLOS ONE’s publication criteria as it currently stands. Therefore, we invite you to submit a revised version of the manuscript that addresses the points raised during the review process.

The paper has been reviewed and some concerns voiced.

We sincerely thank you for the opportunity to improve our manuscript. We are also very grateful to the academic editor for the thoughtful comments, and constructive feedback. We have carefully revised the manuscript and hope that the updated version meets the required criteria.

1. As the study uses two RA concentrations, please provide a rationale for selecting these specific doses and whether they compare to physiological levels of RA in limbal epithelial cells. Were dose-response experiments conducted to determine these concentrations?

We thank the reviewer for this insightful comment. The concentrations of 1 µM and 5 µM all-trans retinoic acid (t-RA) were selected based on previously established protocols involving limbal and corneal epithelial cells. Latta et al. (PLOS One, 2021) applied these same concentrations to investigate RA-responsive gene expression in limbal epithelial cells (1). This is supported by XTT assay data using concentrations of 0 nM, 100 nM, 500 nM, 1 µM, 1.5 µM, and 5 µM, which showed a slight increase in cell viability following 24-hour at-RA treatment, and no significant effect with retinol (1). Additionally, comparable concentration ranges (e.g., 0 µM, 10⁻³ µM, 1 µM) have been used in studies of keratinocyte differentiation and ocular surface models (Kim et al., 2012). Although a formal dose–response pilot study was not performed in the present work, the selected concentrations reflect commonly used doses that are sufficient to activate RA signaling pathways in vitro and are supported by previous literature.

Relatively few studies have quantified retinoic acid (RA) concentrations using highly sensitive analytical techniques. Napoli et al. reported plasma concentrations of total RA in healthy male volunteers ranging from 2.8 to 6.6 ng/mL, with a mean of 4.9 ng/mL—equivalent to approximately 9.3 to 22 nM, based on the molecular weight of all-trans retinoic acid (atRA; ~300 g/mol) (3). Another study documented tissue concentrations of atRA ranging from 20 to 600 nM, varying by tissue type and physiological context (4). However, due to the inherent challenges of direct measurement, there are currently no published data on the precise concentration of RA in limbal epithelial cells under normal physiological conditions in humans. We recognize that the RA concentrations used in our study exceed physiological levels, which are generally estimated to fall within the low nanomolar range in limbal tissues. Nonetheless, supra-physiological concentrations are commonly employed in in vitro experiments to reliably activate RA receptors, thereby improving the detectability of RA-mediated effects in cell culture systems.

In response to the reviewer’s comment, we have clarified this rationale in the revised Materials and Methods Section (lines 198–201) and have added references to the aforementioned studies:

„The RA concentrations used in this study are based on commonly reported doses in the literature that ensure sufficient activation of RA receptors (20,25). Although all-trans retinoic acid (t-RA) was used for the treatment, we will refer to it as RA throughout the manuscript for clarity.”

2. It is unclear what happened to protein expression in control siRNA transfected cells. At least some westerns should cover this gap. The manuscript does not discuss potential off-target effects of siRNA.

We appreciate the reviewer raising this important point. To minimize potential off-target effects, we used a commercially validated PAX6-targeting siRNA and compared it to a scrambled siRNA negative control, as detailed in the Methods (lines 192–194).

In this study, the only variable between the two groups was PAX6 expression, as both were subjected to the same siRNA treatment conditions. Therefore, the observed effects can reasonably be attributed to changes in PAX6 expression rather than off-target effects of the siRNA. This approach—using siRNA knockdown to investigate the role of a single gene—is widely accepted and has been employed in previous studies as well (5,6).

Furthermore, the gene and protein expression changes observed in components of the RA signaling pathway are consistent with previously reported findings in both PAX6 knockdown models and primary aniridia-derived cells, as discussed and summarized in Table 5. We fully acknowledge the importance of distinguishing between direct and off-target effects and have now included a clarifying statement in the Discussion Section (lines 582–588):

„To minimize potential off-target effects of siRNA treatment, we used a commercially validated PAX6-targeting siRNA and compared it to a scrambled siRNA negative control.”

3. Please elaborate on statistical analysis as suggested.

Thank you for this important clarification request. We confirm that Dunnett’s post hoc test, which is specifically designed to control type I error when comparing multiple groups to a single control (in this case, the PAX6 negative control with 0 µM RA treatment), was applied following two-way ANOVA. The p-values reported in the Results reflect these adjustments. Dunnett’s test was recommended by GraphPad Prism 10.0 as the most appropriate method based on our experimental design, in preference to other correction methods (e.g., Bonferroni, Šidák, or Holm-Šídák).

We now clearly state in the Materials and Methods Section (lines 281–283) that adjusted p-values were calculated using Dunnett’s test for multiple comparisons. No additional corrections (e.g., Bonferroni) were applied, as they were not deemed necessary under this analysis framework.

„A two-way ANOVA followed by the Dunnett post hoc test, was used to determine statistical significance.”

4. Was there any effect of siRNA on cell migration (from your or literature data)?

We thank the reviewer for raising this important point. In the present study, our primary focus is on the effects of PAX6 knockdown and RA treatment on gene/protein expression and cell proliferation. As such, we did not include cell migration assays (e.g., scratch/wound healing assays) as part of our experimental design. However, existing literature does suggest a link between PAX6 knockdown and impaired corneal epithelial cell migration. For example, Ramaesh et al. (7) reported defective epithelial wound healing in PAX6+/− mice, which was attributed to abnormalities in epithelial cell migration and differentiation. Additionally, unpublished data from our group indicate a significantly reduced migration rate in siRNA-mediated PAX6 knockdown limbal epithelial cells compared to controls, observed 6 hours post-scratch. While these findings are promising, they remain preliminary and will be explored in greater depth in future studies.

We acknowledge that impaired migration may indeed be a downstream effect of PAX6 knockdown and agree that further investigation using functional assays is warranted. However, given that this was not a central aim of the current study, such analyses have not been included here.

5. Please clarify the issue about RA treatment influence on PAX6 as suggested.

Thank you again for this insightful observation. As noted in our response to Minor Comment 5, PAX6 has been reported as RA-inducible during early eye development, particularly in the formation of the embryonic lens and optic cup (8,9).

However, in our current study using adult human limbal epithelial cells (LECs), we did not observe any significant changes in PAX6 mRNA or protein levels following treatment with 1 µM or 5 µM RA (see Fig. 2). This suggests that the regulatory relationship between RA and PAX6 may be specific to certain developmental stages, tissues, or cell types.

We have now revised the Discussion (lines 582–588) to clarify that, although PAX6 may be RA-inducible in some developmental contexts, this effect was not observed in our in vitro adult LEC model—consistent with previous findings highlighting the cell-type-specific nature of RA responsiveness.

„Although RA-induced PAX6 expression is well documented during early developmental stages, it appears to be highly context-dependent. Our results suggest that in adult LECs, PAX6 is not directly upregulated by RA under the conditions tested.”

6. In the title, please consider adding the word “human”.

We appreciate this helpful suggestion. To more accurately reflect the use of primary human limbal epithelial cells in our model, we have revised the title to:

“Effect of Retinoic Acid Treatment on the Retinoic Acid Signaling Pathway in a Human siRNA-Based Aniridia Limbal Epithelial Cell Model, In Vitro”

This updated title is now reflected on the title page and in the manuscript header (lines 1–2). 

[Comments from Reviewer 1]

Reviewer #1: This study investigates the interplay between retinoic acid (RA) signaling and PAX6 deficiency in limbal epithelial cells (LECs), a model relevant to aniridia-associated keratopathy (AAK). By knocking down PAX6 in primary human LECs and treating with RA, the research reveals significant alterations in gene/protein expression of RA metabolic enzymes (e.g., ALDH1A1, CYP26A1), binding proteins (CRABP2), receptors (RXRA, RARA/B), and downstream targets (VEGF-A, PPARG). Notably, PAX6 knockdown disrupted RA-responsive pathways, while exogenous RA dose-dependently modulated these effectors, particularly suppressing cell proliferation in controls. These findings highlight RA signaling as a potential therapeutic target in AAK pathogenesis by elucidating its dysregulation in PAX6-deficient LECs. However, some concerns need to be addressed properly to strengthen the rigour of these findings.

We would like to sincerely thank Reviewer 1 for the insightful comments.

Major

1. The study uses 1µM and 5µM RA concentrations, but the rationale for selecting these specific doses is not fully explained, the author should justify the RA concentrations used in this study. Since RA is highly potent, were dose-response studies or pilot experiments conducted to determine these concentrations? Additionally, how do these doses compare to physiological levels of RA in limbal epithelial cells?

We thank the reviewer for this insightful comment. The concentrations of 1 µM and 5 µM all-trans retinoic acid (atRA) were selected based on previously established protocols in our laboratory involving limbal and corneal epithelial cells. For instance, Latta et al. (PLOS One, 2021) employed these same concentrations to investigate RA-responsive gene expression in limbal epithelial cells (1). In that study, an XTT assay using a range of concentrations (0 nM, 100 nM, 500 nM, 1 µM, 1.5 µM, and 5 µM) demonstrated a minor increase in cell viability after 24 hours of atRA treatment, with no significant impact observed following retinol treatment. Similar concentration ranges (0 µM, 10⁻³ µM, 1 µM) have also been used in keratinocyte differentiation and ocular surface models (Kim et al., 2012). Although a formal dose-response study was not conducted in the present work, the chosen concentrations are in line with commonly used doses that effectively activate RA signaling in vitro.

Only a few studies have quantified physiological RA concentrations using highly sensitive analytical techniques. Napoli et al. reported plasma concentrations of total RA in healthy male volunteers ranging from 2.8 to 6.6 ng/mL, with a mean of 4.9 ng/mL—equivalent to approximately 9.3 to 22 nM, based on the molecular weight of atRA (~300 g/mol) (3). Additionally, tissue concentrations of atRA have been reported to vary between approximately 20 and 600 nM, depending on tissue type and physiological context (4). However, due to the technical challenges of direct measurement, there are no published data on endogenous RA concentrations specifically in limbal epithelial cells under normal physiological conditions. We acknowledge that the RA concentrations used in our study exceed physiological levels, which are generally estimated to be in the low nanomolar range in limbal tissues. Nevertheless, supra-physiological concentrations are widely employed in vitro to ensure consistent activation of RA receptors, thereby amplifying the detectability of RA-mediated cellular responses.

To address the reviewer’s point, we have now clarified this rationale in the revised Materials and Methods Section (lines 198–201) and added citations to the relevant studies.

„The RA concentrations used in this study are based on commonly reported doses in the literature that ensure sufficient activation of RA receptors (20,25). Although all-trans retinoic acid (t-RA) was used for the treatment, we will refer to it as RA throughout the manuscript for clarity.”

2. The manuscript does not discuss potential off-target effects of siRNA. Can the authors confirm that the observed changes in RA pathway genes are directly due to PAX6 knockdown rather than siRNA transfection artifacts?

We appreciate the reviewer raising this important point. To minimize potential off-target effects, we used a commercially validated PAX6-targeting siRNA and compared it to a scrambled siRNA negative control, as detailed in the Methods section (lines 192–194).In this study, the only difference between the experimental and control groups was the presence of the PAX6-targeting siRNA. Therefore, PAX6 expression is the primary variable, and the observed effects can be attributed specifically to PAX6 knockdown rather than off-target effects of the siRNA treatment. This approach is widely used to investigate the role of individual genes and has been employed in several of our previous studies as well (5,10).

Moreover, the observed changes in gene and protein expression related to RA pathway components were consistent with findings from other PAX6 knockdown models and primary aniridia-derived cells, as discussed and summarized in Table 5.

We fully recognize the importance of distinguishing direct effects from potential off-target effects and have now included a clarifying statement in the Discussion Section (lines 586–588).

„To minimize potential off-target effects of siRNA treatment, we used a commercially validated PAX6-targeting siRNA and compared it to a scrambled siRNA negative control.”

Minor

3. The manuscript states that statistical analysis was performed using two-way ANOVA followed by Dunnett’s test, but p-values for multiple comparisons should be clarified. Were adjustments for multiple testing (e.g., Bonferroni correction) applied?

Thank you for this clarification request. We confirm that Dunnett’s post hoc test, which is specifically designed to control the type I error rate when comparing multiple experimental groups to a single control (PAX6 negative control with 0 µM RA treatment), was applied following the two-way ANOVA. The p-values reported in the Results section reflect these adjustments. Dunnett’s test was recommended as the most appropriate method for our experimental design by GraphPad Prism 10.0, in preference to alternative correction methods (e.g., Bonferroni, Šidák, or Holm-Šídák).

Accordingly, we now clearly state at the Materials and Methods Section (lines 281–283) that adjusted p-values were calculated using Dunnett’s test for multiple comparisons. No additional correction (e.g., Bonferroni) was required under this analysis framework.

„A two-way ANOVA, followed by the Dunnett post hoc test, was used to determine statistical significance.”

4. The term “retinoic acid” is sometimes abbreviated as RA, but in other places, t-RA (all-trans retinoic acid) is used. Please ensure consistency.

We agree with the reviewer that the terminology may cause confusion and appreciate the opportunity to clarify. In this study, the retinoic acid (RA) used is all-trans retinoic acid (t-RA). Throughout the manuscript, we use "RA" as a general term referring specifically to t-RA. However, as noted in the Discussion (lines 498–499), a portion of t-RA can undergo isomerization t

---

## [Decision Letter · Decision Letter 1]

Effect of Retinoic Acid Treatment on the Retinoic Acid Signaling Pathway in a Human siRNA-Based Aniridia Limbal Epithelial Cell Model, In Vitro

PONE-D-25-11947R1

Dear Dr. Hsu,

We’re pleased to inform you that your manuscript has been judged scientifically suitable for publication and will be formally accepted for publication once it meets all outstanding technical requirements.

Kind regards,

Alexander V Ljubimov, Ph.D.

Academic Editor

PLOS ONE

Additional Editor Comments (optional):

Reviewers' comments:

Reviewer's Responses to Questions

**Comments to the Author**

1. If the authors have adequately addressed your comments raised in a previous round of review and you feel that this manuscript is now acceptable for publication, you may indicate that here to bypass the “Comments to the Author” section, enter your conflict of interest statement in the “Confidential to Editor” section, and submit your "Accept" recommendation.

Reviewer #1: All comments have been addressed

2. Is the manuscript technically sound, and do the data support the conclusions?

Reviewer #1: Yes

3. Has the statistical analysis been performed appropriately and rigorously? 

Reviewer #1: Yes

4. Have the authors made all data underlying the findings in their manuscript fully available?

Reviewer #1: (No Response)

5. Is the manuscript presented in an intelligible fashion and written in standard English?

Reviewer #1: Yes

6. Review Comments to the Author

Reviewer #1: (No Response)

7. PLOS authors have the option to publish the peer review history of their article (what does this mean? ). If published, this will include your full peer review and any attached files.

**Do you want your identity to be public for this peer review?** For information about this choice, including consent withdrawal, please see our Privacy Policy .

Reviewer #1: No

---

## [Editor Report · Acceptance letter]

PONE-D-25-11947R1

PLOS ONE

Dear Dr. Hsu,

I'm pleased to inform you that your manuscript has been deemed suitable for publication in PLOS ONE. Congratulations! Your manuscript is now being handed over to our production team.

Kind regards,

on behalf of

Dr. Alexander V Ljubimov

Academic Editor

PLOS ONE